# Determinants of Higher Mortality at Six Months in Patients with Hip Fracture: A Retrospective Study

**DOI:** 10.3390/jcm11092514

**Published:** 2022-04-29

**Authors:** Enrique González-Marcos, Enrique González-García, Paula Rodríguez-Fernández, Esteban Sánchez-González, Jerónimo J. González-Bernal, Josefa González-Santos

**Affiliations:** 1RACA 11 Artillery Regiment, Cid Campeador Military Base, 09193 Burgos, Spain; enriquegojs@gmail.com; 2Traumatology and Orthopedic Surgery Service, Burgos University Hospital, 09006 Burgos, Spain; enriqueglezgar@yahoo.es; 3Department of Health Sciences, University of Burgos, 09001 Burgos, Spain; mjgonzalez@ubu.es; 4Department of Health Sciences, University of Jan Kochanowski, 25-369 Kielce, Poland; estebansg2001@gmail.com

**Keywords:** hip fracture, mortality, associated factors, elderly

## Abstract

(1) Background: Hip fracture is a pathology with high mortality, but the lack of a universal adaptation of the factors associated with death makes it difficult to predict risk and implement prevention in this group. This study aimed to identify the factors that determine a higher mortality at six months following hip fracture. (2) Methods: A retrospective longitudinal study, whose study population consisted of patients over 65 years of age. The main variable was mortality at 6 months of fracture. Relevant data related to sociodemographic and clinical variables for subsequent bivariate (χ^2^) and multivariate analysis were obtained. (3) Results: In all, 665 people participated in the study, 128 of whom died within 6 months of the fracture. The multivariate adjusted analysis demonstrated significant relationships between the main variable and aspects such as institutionalization at discharge (Odds Ratio (OR) = 2.501), a worse overall functional capacity (OR = 2.453) and cognitive capacity (OR = 3.040) at admission, and complications such as heart failure (OR = 5.767) or respiratory infection (OR = 5.308), in addition to the taking of certain drugs and the presence of a greater number of comorbidities. (4) Conclusions: There are certain factors related to higher mortality at six months in patients with hip fracture who are aged 65 years or older.

## 1. Introduction

Hip fracture is a pathology with a high mortality and incidence, which has increased by 1.5% every year during the last decade in Spain, currently producing 104 cases per 100,000 inhabitants, which means about 50,000 fractures per year [1].

A recent study by Blanco-Rubio [2] estimated an early mortality due to hip fracture in Spain of 22%. Globally, the previous knowledge on mortality associated with hip fracture is well established. Hip fracture is a cause of mortality in the elderly, and between 2% and 7% of the world population with this pathology die on admission and almost 30% die a year after suffering it [3], with the male gender and a higher age being added risk factors. The relative risk of death of people with hip fracture is two to three times higher than that of the population of the same age [4], especially in the first six months [5]. Males have demonstrated higher mortality, especially during admission, which has been linked to the so-called “Fragility Index” (FI), which represents the proportion of deficits present in an individual over the total number of age-related health variables considered [6,7,8]. Hip fracture, and fractures in general, is accepted as a fragility fracture when it occurs as the result of a low-energy mechanism, such as a fall at the same level [9].

In Spain, several studies have been performed to know the factors associated with hip fracture. The factors found in recent studies have been many and varied, but most research considers variables such as age, patient dependence [10], and comorbidities [11]. Likewise, mortality in people with hip fracture, even at two years, has been shown to be significantly related to factors such as respiratory infection, decompensated heart diseases, or dementia [12]; as well as with male gender and complications during admission such as chronic renal failure, anemia, delirium, renal functional decompensation, and waiting for surgery more than 2 days [13]. For his part, Blanco-Rubio [2] with an excellent prospective design and a multivariate analysis, related mortality during the first six months with the presence of a heart disease when admitted for hip fracture.

Excess mortality in people with hip fracture derives from the presence of certain factors, which differ markedly from each other in the literature reviewed. This means that there is no universal adaptation of them that facilitates the prediction of risk, complications, functional impairment, or even mortality in people with this problem. Therefore, the main objective of this research was to identify the factors that determine a higher mortality at six months following hip fracture in a sample of patients over 65 years of age who attend the University Hospital of Burgos (HUBU), Spain.

## 2. Materials and Methods

### 2.1. Study Design—Participants

A retrospective longitudinal study was designed with the following inclusion criteria:
−Patients aged 65 years or older;−Who suffered a hip fracture by a low-energy mechanism; −In the biennium 14 March 2019–14 March 2021, all patients admitted to the HUBU with these characteristics were included in the study and followed after discharge from the outpatient clinics of the Orthopedic Surgery and Traumatology Service of the same hospital through face-to-face and non-face-to-face consultations.

The exclusion criteria were as follows:
−Patients with peri-prosthetic fractures; −Peri-synthesis fractures; −Pathological fractures, that is, on bones affected by primary tumor or metastasis, were excluded from the study; −Likewise, patients who were referred to other hospitals without completing the treatment or follow-up period for any cause, except death were also excluded.

### 2.2. Sample Size 

The sample size was estimated following the procedure for finite populations, using the formula $n=\frac{N*{\left(Z_{\alpha }=1.96\right)}^{2}*p*q}{\delta^{2}*\left(N-1\right)+{\left(1.96\right)}^{2}*p*q}$. This calculation took into account the known population reported by the National Institute of Statistics [14] and the results of a similar study [15], establishing a proportion of hip fractures in the population of 0.389% (*p* = 0.000398, and its complementary q = 0.99602) and assuming a sampling error of 1% ($\delta^{2}$ = 01 in the formula). Based on this, it was concluded that the sample should consist of 152 patients with hip fractures under care by the HUBU.

### 2.3. Main Outcomes—Instruments

The head of the Traumatology Section of the Orthopedic Surgery and Traumatology Service was responsible for collecting the data for subsequent analysis, obtained through the electronic clinical history of each participant. Information related to sociodemographic and general clinical characteristics was considered by variables such as sex (female or male); age; original place of residence and at discharge (home or residence); type of treatment (conservative or surgical); type of fracture (intracapsular or extracapsular); surgical risk (assessed according to the American Society of Anesthesiologists’ physical status classification (ASA)) [16]; surgical delay (understood as the days elapsed from admission to intervention); length of hospital stay or days between admission and discharge; cognitive impairment before and after admission (assessed using the Pfeiffer Scale (PS)) [17]; functional capacity (assessed by the Barthel Index (BI)) [18]; and the capacity for standing, sedation, and walking prior, during, and after admission. Comorbidities were also obtained (active oncological process, chronic anemia, atrial fibrillation, heart failure, valvular diseases, ischemic heart disease, chronic obstructive pulmonary disease (COPD), and chronic renal failure), as were relevant drugs (anti-hypertensives, antiplatelet agents, anticoagulants, Sintrom, neuroleptics, bronchodilators, oxygen at home, and protein supplements and thickeners) and complications of admission such as the need for a transfusion, constipation, “delirium”, deterioration of kidney function, or ulcers.

The evolution of the patients and/or their death were recorded during the 6 months after the fracture in the face-to-face or telematic check-ups.

### 2.4. Statistical Analysis

To characterize the sample, absolute frequencies and percentages were used, if the variables were categorical, or mean and standard deviation (SD) were employed in the case of continuous variables. The categorical variables of more than two categories were dichotomized based on previous studies and the continuous variables based on the average score in order to obtain two groups as homogeneous as possible.

Bivariate analyses were performed to study the relationship between dichotomous variables and death at 6 months using the Pearson independence test (χ^2^) as well as the likelihood ratio. In the comparisons of significant dichotomous variables, the ratio of advantages or odds ratio (OR) was also obtained.

To quantify the magnitude of these relationships and identify possible predictive factors of mortality at 6 months based on the independent variables, a binary logistic regression was performed taking the fact of dying or surviving the sixth month of the hip fracture as a dichotomous dependent variable. The multivariate analysis was adjusted for age (≥85 years), sex (male), surgical delay (≥3 days), and length of hospital stay (≥11 days). All variables with a value of *p* < 0.05 in the bivariate analysis were included as independent variables in the multivariate analysis, obtaining a statistic (χ^2^ Wald), a *p* value, and a risk measured in adjusted OR for each of them.

Statistical analysis was performed with SPSS software version 25 (IBM-Inc., Chicago, IL, USA). For the analysis of statistical significance, a *p*-value < 0.05 was established.

## 3. Results

The study sample consisted of a total of 665 people, 128 of whom died during the 6 months after the hip fracture. The age of the participants was between 65 and 102 years, with a mean of 86.2 years with 76.7% women (n = 510) and 23.3% men (n = 155). Males demonstrated a significantly higher mortality at six months compared to women, although the advantage ratio was only moderately higher (OR = 1.59). The survival of people aged 85 and over was significantly lower, with a risk of dying at 6 months that was 4426 times higher than that for younger people. Other sociodemographic factors such as pre- and post-fracture institutionalization also appear to be a risk factor for death at 6 months following hip fracture (Table 1).

Regarding the type of intervention and fracture, there was a significant relationship between surgery or conservative treatment, patients who received conservative treatment reported high mortality (OR = 8.985) compared to those who received surgical treatment (OR = 0.111), but no statistically significant differences were found in the mortality at 6 months of patients with intracapsular and extracapsular hip fractures (Table 1). Likewise, patients with a higher surgical risk demonstrated more than twice the chances (OR = 2.308) of dying at six months after fracture than those classified as ASA I or II. A surgical delay of three or more days (the average was 4.58 ± 3.79 days) tripled (OR = 3.352) the significant risk of dying in the sixth month compared with those operated on during the first 48 h of admission. The same happened with the hospital stay (10.46 ± 5.44), with patients with a stay equal to or greater than eleven days were associated with a possibility of dying at the sixth month that was 2.438 greater than that for those who were hospitalized for less than eleven days.

Regarding the clinical characteristics related to the ability to stand and sit and the gait, statistically significant relationships were found in all the variables studied, especially in the results during admission, since those patients who were not able to stand or walk demonstrated more than ten times more chances of dying in the sixth month of having fractured their hip than those who did manage to stand and walk, and those who did not manage to sit during admission showed 60.59 times more chances of dying than those who did sit (Table 1).

Table 2 shows variables related to comorbidities and drugs prior to admission, as well as some new prescriptions after discharge. All the previous comorbidities recorded in the clinical history demonstrated a significant relationship with mortality at six months after hip fracture, highlighting suffering from active oncological process, chronic anemia, and heart failure, with a probability ratio of OR = 3.109, OR = 3.457, and OR = 3.510, respectively. Likewise, a strong association was found between mortality at six months and the presence of three or more of the previously mentioned comorbidities, since having three or more pathologies prior to admission quintupled the possibility of death at the sixth month after the hip fracture (OR = 5.034), and according to Kaplan–Meier’s analysis, the survival of these patients seemed to be very decreased (log Rank = 20.62, *p* < 0.001).

Regarding the drugs prior to admission, no mortality relationship was found with the previous taking of antiplatelet agents, but there was an association with the rest of the drugs studied. In particular, the use of bronchodilators (OR = 2.429) and home oxygen (OR = 4.959) prior to admission were associated with a higher chance of dying at the sixth month following hip fracture in patients who used it compared to those who did not (Table 2). When the 30-day mortality was extracted from the series, the drugs with a significant relationship with the increase in mortality were antihypertensives and home oxygen prior to admission, which meant an increase in the chances of early death (OR = 2.154 and OR = 0.269, respectively). Patients prescribed protein supplements (OR = 12.019) and thickeners (OR = 8.429) “de novo” at discharge also demonstrated significantly higher mortality at six months following hip fracture compared to those who did not receive such a prescription (Table 2).

Regarding the complications presented during hospital admission in patients with hip fracture, all those studied were related to a higher mortality in the sixth month. Among all the complications, the infection of the surgical wound stands out, since its presence implies that the possibility of death at the sixth month is almost eight times greater (OR = 7.790), followed by respiratory infection (OR = 9.550) and acute heart failure (OR = 10.350). In addition, patients with impaired renal function and/or pressure ulcers on admission showed an almost a five times greater chance of dying within six months of the fracture (OR = 4.925 and OR = 4.955, respectively). The “delirium” on admission of the elderly patient with a fractured hip, which is a very frequent event, was also shown to double the possibility of death at the sixth month (OR = 2.689) (Table 3).

Below are the results of binary logistic regression to estimate the relationship between mortality at the sixth month following hip fracture and relevant sociodemographic and clinical characteristics (Table 4). 

Furthermore, some drugs such as Sintrom and antiplatelet agents and comorbidities such as valvular diseases, ischemic heart disease, and atrial fibrillation were not related to mortality at the sixth month in patients with hip fracture (Table 5). 

Table 6 shows the multivariate analysis adjusted for age, sex, surgical delay, and hospital stay to estimate mortality at the sixth month following hip fracture and characteristics and complications at admission. It was found that the only characteristics and complications in admission related to mortality at the sixth month were acute heart failure, respiratory infection, and deterioration of renal function (OR = 5.767, OR = 5.308, and OR = 3.622, respectively). Anemia, delirium, and transfusion during admission were also significant variables in logistic regression.

## 4. Discussion

The results of this research show that there are factors related to higher mortality at the sixth month following hip fracture. Sociodemographic characteristics such as sex, age, length of hospital stay, or place of residence before and after discharge have been shown to be significantly related to the main variable. Almost all of the literature reviewed states that age is the factor that is always related to the highest mortality in patients with hip fractures [13,15,19,20,21]. Moreover, the male sex is described to have a lower significance on the relationship [13,15,19,20,21,22,23,24,25], since the comorbidities presented tend to influence mortality to a greater extent [26]. Rapp et al. [27] found excess mortality in the institutionalized elderly population with hip fracture regardless of sex during the three and six months after the episode, which decreases from the sixth month on. In this sense, our study points out that institutionalization at hospital discharge has an important relationship with mortality at six months, especially in the adjusted analysis, which coincides with the findings of Cree et al. [25], who in prospective work to study factors related to mortality at three months and institutionalization after hip fracture found that age, male sex, and cognitive impairment were the factors that determined a greater need for institutionalization after discharge and also higher mortality.

General clinical factors such as cognitive impairment, functional characteristics (BI), and surgical risk (ASA) have also shown a relevant role in mortality at the sixth month of patients with hip fracture studied in this research. The literature relates cognitive impairment to high mortality in people with hip fracture [13,23,25,28,29,30,31,32] and is usually correlated with lower postoperative ambulatory capacity [33] and a consequent higher mortality [33,34]. These results coincide with the findings of this research, since a worse capacity for global functionality, gait, and sedation was related to a higher mortality at the sixth month of the hip fracture. In this line, Uriz-Otano et al. [23] found a relationship between the mean BI value at admission and 3-year mortality, and Folbert et al. [24] and Duaso et al. [13] found in their studies that hip fracture patients who died one year after the episode had the lowest BI values at admission and discharge. According to Aranguren et al. [10], mortality at one and two years in patients with a BI at admission ≤60 is significantly higher, as shown by the results of this research at six months. It has also been shown that survival in nonagenarian patients with hip fracture who are unable to walk is significantly lower than that of those who do manage to walk [35], and in line with our results, Heinonen et al. [36] described that the inability to sit, stand, or walk in the first two weeks after hip fracture is the main factor related to an increase of mortality per year in people with hip fractures aged 65 years or older.

In terms of treatment, not operating has a very close relationship with the mortality of these patients [37]. Like our results, the literature associates surgical-anesthetic risk (ASA) with mortality [28,38,39,40], and one study considers it the most important factor determining mortality at two years, although lower than age and cognitive decline [31]. In the case of surgical treatment, a delay of more than 48 h has been shown to be a determinant of mortality with high scientific evidence [41,42,43,44,45], which coincides with the findings of the present research. It is worth mentioning a study carried out by Uzoigwe et al. [46], where the time limit was set to relate surgical delay to mortality not at 48 h, but at 36.

As in our results, mortality is not usually associated with the type of fracture [47], but there are studies that show a higher mortality of extracapsular fractures compared to intracapsular fractures even after adjusting for the variables age and comorbidities [48,49]. The latter influence the mortality of patients with hip fracture [50,51,52], and the relationship between plurimorbidity and mortality from this pathology is generally established [53,54]. In this research, mortality was significantly related to the presence of three or more pathologies before the fracture occurs. 

As for drugs that are prescribed before admission, a recent study demonstrates the relationship with 30-day mortality with the use of antihypertensives and psychotropic drugs [55], and Uriz-Otano et al. [23] found a significant relationship with previous taking of benzodiazepines and neuroleptics and mortality at 3 years. In this line, our results show a significant relationship between mortality at the sixth month following hip fracture and previous drugs, except for antiplatelet drugs. In his work, Wordsworth et al. [56] also found no significant relationship between mortality and taking antiplatelet drugs. Likewise, there is evidence that systematically giving oral protein supplements to all elderly hip operated patients reduces complications and mortality at 1 year [57]. In our research, on the other hand, patients taking protein supplements or thickeners at discharge demonstrated a significantly higher probability of dying within six months of the fracture, which may be due to the low representativeness of the sample since these supplements were prescribed in less than 20 patients.

Finally, a more significant relationship of mortality with complications at admission than with previous comorbidities has been found, as in the studies of other authors [58]. This fact is underlined because mortality at six months after hip fracture has been shown to be six times higher in patients suffering complications at admission [59,60]. In agreement with our results, acute renal failure and the need for transfusion [61], acute heart failure [62,63], respiratory infection [64], and agitation and disorientation syndrome [65] are complications of admission significantly related to the highest mortality at six months; although a recent Dutch study denies that “delirium” is associated with higher mortality [66]. Additionally, in agreement with our results, the study conducted by Bielza et al. [67] found that, although the complications that appeared in patients with hip fracture are very numerous, delirium, acute urinary retention, acute heart failure, acute respiratory infection, and deterioration of renal function acquire special relevance. A recent Spanish study identified pneumonia, cardiocirculatory disorders, and delirium as the main determinants of death at two years following hip fracture, a risk that increased with age and male sex as in this research [12].

This study should be considered in the context of its limitations. Despite having collected a wide variety of clinical and sociodemographic data, there were no complicating events that could influence the mortality of patients with hip fracture beyond hospital discharge. The exact cause of death of the deceased patients was also not recorded, which could have allowed us to better adjust the influence of fracture morbidity. No objectiv mortality or comorbidity assessment scale was applied, such as the usually used Charlson scale [52], nor were other, also objective scales, such as the “Geriatric Comorbidity Index” [68], the “Cumulative Illness Rating Scale” [69], or the “Index of Coexisting Diseases” [70] or scales of gait assessment such as the so-called “Functional Ambulation Classification (FAC)” used [71]. However, although they have a wide use in the literature, their use in it is heterogeneous and not completely validated to the specific case of the elderly person affected by hip fracture in their environment. The strengths of this publication are the great representativeness of the sample in the health area studied, where the vast majority of elderly patients with hip fracture are treated at the HUBU, covering both the rural and urban populations. In addition, practically all the risk factors that can influence hip fracture mortality in the elderly were collected, taking into account the previously reviewed literature. It should also be noted that the temporal evolution to six months, together with the multivariate study, allows us to affirm an etiopathogenic relationship of the factors studied and the death at the sixth month in patients with hip fracture.

## 5. Conclusions

There are certain factors related to higher mortality at six months in patients with hip fracture aged 65 years or older. The main characteristics related to death at the sixth month were having a worse global functional and cognitive capacity at admission and discharge, an increased surgical risk, institutionalization at discharge, a greater number of comorbidities, and the appearance of complications during admission.

## Figures and Tables

**Table 1 jcm-11-02514-t001:** Chi^2^ test results between mortality at sixth month and relevant sociodemographic and clinical characteristics.

Sociodemographic and General Clinical Characteristics	Death at 6th Month		Chi^2^ Test		OR		
	Yes	No	χ^2^ χ RV^2^	*p*-Value	Rho	Lower Limit	Upper Limit
**Sex**							
Female (n = 510)	89 (17.5%)	421 (82.5%)	4.06	0.044	0.629	0.410	0.965
Male (n = 155)	39 (25.2%)	116 (74.8%)	4.34	0.037	1.590	1.036	2.442
**Age**							
≥85 years (n = 441)	112 (25.4%)	329 (74.6%)	30.68	<0.001	4.426	2.549	7.685
65 to 84 years (n = 224)	16 (7.1%)	208 (92.9%)	36.37	<0.001	0.226	0.130	0.392
**Previous place of residence**							
Institutionalized (n = 197)	51 (25.9%)	146 (74.1%)	7.35	0.007	1.774	1.187	2.651
At home (n = 468)	77 (16.5%)	391 (83.5%)	7.62	0.006	0.564	0.377	0.842
**Place of residence upon discharge**							
Institutionalized (n = 294)	65 (22.1%)	229 (77.9%)	23.64	<0.001	3.238	2.003	5.234
At home (n = 335)	27 (8.1%)	308 (91.9%)	25.17	<0.001	0.309	0.191	0.499
**ASA surgical risk**							
ASA I + II (n = 296)	31 (10.5%)	265 (89.5%)	12.67	<0.001	0.433	0.275	0.683
ASA III + IV (n = 334)	71 (21.3%)	263 (78.7%)	13.83	<0.001	2.308	1.464	3.638
**Type of treatment**							
Conservative (n = 26)	17 (65.4%)	9 (34.6%)	34.03	<0.001	8.985	3.904	20.677
Surgical (n = 639)	111 (17.4%)	528 (82.6%)	27.82	<0.001	0.111	0.048	0.256
**Surgical delay**							
≤2 days (n = 151)	10 (6.6%)	141 (93.4%)	12.49	<0.001	0.298	0.151	0.589
≥3 days (n = 479)	92 (19.2%)	387 (80.8%)	15.67	<0.001	3.352	1.697	6.620
**Length of hospital stay**							
≤11 days (n = 402)	58 (14.4%)	344 (85.6%)	14.42	<0.001	0.465	0.315	0.687
>11 days (n = 263)	70 (26.6%)	193 (73.4%)	14.90	<0.001	2.151	1.456	3.178
**Type of fracture**							
Intracapsular (n = 274)	49 (17.9%)	225 (82.1%)	274	0.42	*p* > 0.05		
Extracapsular (n = 391)	79 (20.2%)	312 (79.8%)	391	0.56			
**PS on entry**							
No impairment or mild (n = 500)	86 (17.2%)	414 (82.8%)	4.92	0.027	0.608	0.400	0.926
Moderate to severe (n = 165)	42 (25.5%)	123 (74.5%)	5.19	0.023	1.644	1.080	2.503
**BI on entry**							
≤60 (n = 113)	42 (37.2%)	71 (62.8%)	26.75	<0.001	3.205	2.053	5.005
>60 (n = 552)	86 (15.6%)	466 (84.4%)	24.68	<0.001	0.312	0.200	0.487
**Ambulation at admission**							
Capablen (n = 587)	92 (15.9%)	486 (84.1%)	29.92	<0.001	0.268	6.524	16.184
Incapable or with difficulty (n = 87)	36 (41.4%)	51 (58.6%)	26.76	<0.001	3.729	0.062	0.153
**Standing and walking during admission**							
Capable (n = 114)	65 (57%)	49 (43%)	123.36	<0.001	0.097	0.062	0.153
Incapable (n = 511)	63 (11.4%)	488 (88.6%)	103.90	<0.001	10.28	6.524	16.184
**Seating during admission**							
No (n = 14)	13 (92.9%)	1 (7.1%)	45.13	<0.001	60.59	7.848	467.80
Yes (n = 651)	115 (17.7%)	536 (82.3%)	37.14	<0.001	0.017	0.002	0.127
**Ambulation at discharge**							
Capable (n = 364)	22 (6%)	342 (94%)	49.35	<0.001	0.179	0.108	0.299
Unable (n = 265)	70 (26.4%)	195 (73.6%)	51.44	<0.001	5.580	3.350	9.296
**PS at discharge**							
No impairment or mild (n = 468)	57 (12.2%)	411 (87.8%)	8.02	0.005	0.499	0.313	0.795
Moderate to severe (n = 161)	35 (21.7%)	126 (78.3%)	8.18	0.004	2.003	1.257	3.191

OR: odds ratio; ASA: American Society of Anaesthesiologists’ physical status classification; PS: Pfeiffer Scale; BI: Barthel Index.

**Table 2 jcm-11-02514-t002:** Results of the Chi^2^ test between mortality at the sixth month and comorbidities and drugs at admission.

Comorbidities and Drugs on Admission	Death at 6th Month		Chi^2^ Test		OR		
	Yes	No	χ^2^ χ RV^2^	*p*-Value	Rho	Lower Limit	Upper Limit
**Active oncological process**							
Yes (n = 84)	32 (38.1%)	52 (61.9%)	20.61	<0.001	3.109	1.901	5.084
No (n = 581)	96 (16.5%)	485 (83.5%)	18.93	<0.001	0.322	0.197	0.526
**Chronic anemia**							
Yes (n = 134)	50 (37.3%)	84 (62.7%)	33.80	<0.001	3.457	2.261	5.286
No (n = 531)	78 (14.7%)	453 (85.3%)	31.24	<0.001	0.289	0.189	0.442
**Atrial fibrillation**							
Yes (n = 160)	45 (28.1%)	115 (71.9%)	9.94	0.002	1.990	1.311	3.020
No (n = 505)	83 (16.4%)	422 (83.6%)	10.02	0.002	0.503	0.331	0.763
**Heart failure**							
Yes (n = 187)	65 (34.8%)	122 (65.2%)	38.89	<0.001	3.510	2.350	5.241
No (n = 478)	63 (13.2%)	415 (86.8%)	37.21	<0.001	0.285	0.191	0.426
**Valvular heart disease**							
Yes (n = 66)	20 (30). 3%)	46 (69.7%)	5.00	0.025	1.977	1.124	3.477
No (n = 599)	108 (18%)	491 (82%)	5.19	0.023	0.506	0.288	0.890
**Ischemic heart disease**							
Yes (n = 59)	19 (32.2%)	40 (67.8%)	6.11	0.013	2.166	1.208	3.884
No (n = 606)	109 (18%)	497 (82%)	6.20	0.013	0.462	0.257	0.828
**COPD**							
Yes (n = 94)	31 (33%)	63 (67%)	12.27	<0.001	2.405	1.484	3.895
No (n = 571)	97 (17%)	474 (83%)	11.84	0.001	0.416	0.257	0.674
**Chronic renal failure**							
Yes (n = 148)	47 (31.8%)	101 (68.2%)	18.14	<0.001	2.505	1.647	3.810
No (n = 571)	81 (15.7%)	436 (84.3%)	17.56	<0.001	0.399	0.262	0.607
**Antihypertensive**							
Yes (n = 400)	90 (22.5%)	310 (77.5%)	6.31	0.012	1.734	1.144	2.629
No (n = 265)	38 (14.3%)	227 (85.7%)	7.03	0.008	0.577	0.380	0.874
**Antiplatelet agents**							
Yes (n = 104)	21 (20.2%)	83 (79.8%)	0.02	0.896	*p* > 0.05		
No (n = 561)	107 (19.1%)	454 (80.9%)	0.07	0.791			
**Anticoagulants**							
Yes (n = 150)	41 (27.3%)	109 (72.7%)	7.49	0.006	1.850	1.208	2.835
No (n = 515)	87 (16.9%)	428 (83.1%)	7.66	0.006	0.540	0.353	0.828
**Sintrom**							
Yes (n = 80)	26 (32.5%)	54 (67.5%)	9.33	0.002	2.280	1.363	3.813
No (n = 585)	102 (17.4%)	483 (82.6%)	9.15	0.002	0.439	0.262	0.734
**Neuroleptics**							
Yes (n = 115)	30 (26.1%)	85 (73.9%)	3.67	0.055	1.628	1.017	2.605
No (n = 550)	98 (17.8%)	452 (82.2%)	3.94	0.047	0.614	0.384	0.983
**Bronchodilators**							
Yes (n = 77)	26 (33.8%)	51 (66.2%)	10.78	0.001	2.429	1.447	4.079
No (n = 588)	102 (17.3%)	486 (82.7%)	10.41	0.001	0.412	0.245	0.691
**O_2_ at the previous address**							
Yes (n = 29)	15 (51.7%)	14 (48.3%)	18.45	<0.001	4.959	2.328	10.563
No (n = 636)	113 (17.8%)	523 (82.2%)	16.17	<0.001	0.202	0.095	0.430
**Protein supplements “de novo”**							
Yes (n = 17)	11 (64.7%)	6 (35.3%)	31.09	<0.001	12.019	4.326	33.391
No (n = 612)	81 (13.2%)	531 (86.8%)	23.09	<0.001	0.083	0.030	0.231
**Thickeners “de novo”**							
Yes (n = 14)	8 (57.1%)	6 (42.9%)	17.39	<0.001	8.429	2.853	24.900
No (n = 615)	84 (13.7%)	531 (86.3%)	14.00	<0.001	0.119	0.040	0.350

OR: odds ratio; COPD: chronic obstructive pulmonary disease.

**Table 3 jcm-11-02514-t003:** Results of the Chi^2^ test between mortality at the sixth month and characteristics and complications at admission.

Characteristics and Complications at Admission	Death at 6th Month		Chi^2^ Test		OR		
	Yes	No	χ^2^ χ RV^2^	*p*-Value	Rho	Lower Limit	Upper Limit
**Significant anemia**							
Yes (n = 468)	102 (21.8%)	366 (78.2%)	6.05 6.98	0.014 0.008	1.833	1.149	2.925
No (n = 197)	26 (13.2%)	171 (86.8%)			0.546	0.342	0.871
**Transfusion**							
Yes (n = 330)	76 (23%)	254 (77%)	6.06	0.014	1.628	1.101	2.408
No (n = 335)	52 (15.5%)	283 (84.5%)	5.56	0.018	0.614	0.415	0.908
**Delirium**							
Yes (n = 241)	71 (29.5%)	170 (70.5%)	24.34	<0.001	2.689	1.815	3.984
No (n = 424)	57 (13.4%)	367 (86.6%)	24.50	<0.001	0.372	0.251	0.551
**Constipation**							
Yes (n = 294)	68 (23.1%)	226 (76.9%)	4.67	0.031	1.560	1.059	2.297
No (n = 371)	60 (16.2%)	311 (83.8%)	5.08	0.024	0.641	0.435	0.944
**Impaired kidney function**							
Yes (n = 203)	77 (37.9%)	126 (62.1%)	63.90	<0.001	4.925	3.281	7.393
No (n = 462)	51 (11%)	411 (89%)	61.03	<0.001	0.203	0.135	0.305
**Acute heart failure**							
Yes (n = 121)	68 (56.2%)	53 (43.8%)	127.04	<0.001	10.35	6.611	16.203
No (n = 544)	60 (11%)	484 (89%)	107.88	<0.001	0.097	0.062	0.151
**Respiratory infection (pneumonia)**							
Yes (n = 90)	53 (58.9%)	37 (41.1%)	102.30	<0.001	9.550	5.880	15.510
No (n = 575)	75 (13%)	500 (87%)	84.23	<0.001	0.105	0.064	0.170
**Acute urinary retention**							
Yes (n = 75)	22 (29.3%)	53 (70.7%)	4.82	0.028	1.895	1.105	3.251
No (n = 590)	106 (18%)	484 (82%)	5.03	0.025	0.528	0.308	0.905
**Surgical wound infection**							
Yes (n = 5)	3 (60%)	2 (40%)	4.25	0.039	7.970	1.315	48.312
No (n = 625)	99 (15.8%)	526 (84.2%)	4.96	0.026	0.125	0.021	0.761
**Surgical wound seroma**							
Yes (n = 47)	13 (27.7%)	34 (72.3%)	4.05	0.044	2.122	1.078	4.180
No (n = 583)	89 (15.3%)	494 (84.7%)	4.29	0.038	0.471	0.239	0.928
**Pressure ulcers**							
Yes (n = 21)	11 (52.4%)	10 (47.6%)	13.19	<0.001	4.955	2.056	11.939
No (n = 644)	117 (18.2%)	527 (81.8%)	11.95	0.001	0.202	0.084	0.486

OR: odds ratio.

**Table 4 jcm-11-02514-t004:** Results of binary logistic regression to estimate the relationship between mortality at the sixth month and relevant sociodemographic and clinical characteristics.

Sociodemographic and Clinical Characteristics	χ^2^ Wald	*p*-Value	Rho
BI at admission: ≤60, severe or total dependence	10.661	0.001	2.453
PS on admission: moderate to severe impairment	8.310	0.003	3.040
ASA surgical risk: ASA III + IV	3.835	0.050	1.618
Discharge PS: moderate to severe impairment	16.256	<0.001	1.203
Ambulation at admission: unable	11.811	<0.001	2.803
Standing and walking during admission: unable to cope	2.107	0.146	1.510
Previous place of residence: institutionalized	3.493	0.061	1.577
Place of subsequent residence: institutionalized	11.590	<0.001	2.501

ASA: American Society of Anesthesiologists’ physical status classification; PS: Pfeiffer Scale; BI: Barthel Index. Covariates: age, sex, delay ≥3 days, and stay ≥11 days.

**Table 5 jcm-11-02514-t005:** Results of binary logistic regression to estimate the relationship between mortality at sixth month and comorbidities and drugs at admission.

Comorbilities and Drugs on Admission	χ^2^ Wald	*p*-Value	Rho
De novo protein supplements: YES	15.744	<0.001	10.222
Thickeners “de novo”: YES	10.743	0.001	8.303
O_2_ at the previous address: YES	15.640	<0.001	6.186
Active oncological process: YES	16.201	<0.001	3.273
Chronic anemia: YES	18.407	<0.001	2.895
Heart failure: YES	13.222	<0.001	2.360
COPD: YES	6.073	0.013	2.036
Bronchodilators: YES	4.879	0.027	2.004
Anticoagulants: YES	4.879	0.027	2.004
Neuroleptics: YES	5.890	0.015	1.924
Chronic renal failure: YES	6.986	0.008	1.914
Sintrom: YES	3.259	0.071	1.718
Valvular heart disease: YES	1.810	0.178	1.570
Ischemic heart disease: YES	0.743	0.388	1.371
Atrial fibrillation: YES	0.843	0.358	1.259

COPD: chronic obstructive pulmonary disease. Covariates: age, sex, delay ≥3 days, and stay ≥11 days.

**Table 6 jcm-11-02514-t006:** Results of binary logistic regression to estimate the relationship between mortality at the sixth month and characteristics and complications at admission.

Characteristics and Complications at Admission	χ^2^ Wald	*p*-Value	RHO
Surgical wound infection: YES	3.204	0.073	6.124
Acute heart failure: YES	41.915	<0.001	5.767
Respiratory infection: YES	33.931	<0.001	5.308
Impaired kidney function: YES	27.437	<0.001	3.622
Pressure ulcers: YES	1.930	0.164	2.201
Significant anemia: YES	5.818	0.015	2.139
Delirium: YES	9.845	0.001	2.090
Surgical wound seroma: YES	2.273	0.136	1.749
Transfusion: YES	4.595	0.032	1.691
Acute urinary retention: YES	2.684	0.101	1.670
Constipation: YES	1.217	0.269	1.298

Covariates: age, sex, delay ≥3 days, and stay ≥11 days.

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
