# Peer review of "Determinants of Higher Mortality at Six Months in Patients with Hip Fracture: A Retrospective Study"

_jcm, 2022, doi:10.3390/jcm11092514_

Round 1
Reviewer 1 Report
The questions I pointed out are well corrected and answered.
I have no further questions.
Author Response
Response: Thank you very much

Reviewer 2 Report
The study show no signifiant diferences of mortality betwewn surgery or conservative treatment. For this reson is necessary to make a redesign of the study. The study must contain inclusion and exclusion criteria. After the new analysis, the result should be compared, point by point with the other signifiant reports.
Author Response
Review 2
The study show no signifiant diferences of mortality betwewn surgery or conservative treatment. For this reson is necessary to make a redesign of the study.
Response: Thank you very much for pointing it out.
There is a significant relationship between surgery or conservative treatment, indicated in Table 1, and modified in the text. the chi square test cannot indicate significant differences, only a significant relationship. Likewise, the positive value of the odd ratio is indicated. For all these reasons, we believe that it is not necessary to make a redesign of the study, since it is retrospective.
“Regarding the type of intervention and fracture, there is a significant relationship between surgery or conservative treatment, patients who received conservative treatment reported high mortality (OR=8.985) compared to surgical (OR=0.111),”
The study must contain inclusion and exclusion criteria.
Response: Thank you very much for pointing it out. The inclusion and exclusion criteria, already indicated, have been made explicit:
A retrospective longitudinal study was designed with the followings inclusion criteria:
- Patients aged 65 years or older
- Who by a low energy mechanism suffered a hip fracture
- In the biennium 14/03/2019 - 14/03/2021. All patients admitted to the HUBU with these characteristics were included in the study, followed after discharge from the outpatient clinics of the Orthopedic Surgery and Traumatology Service of the same hospital through face-to-face and non-face-to-face consultations.
And exclusion criteria:
- Patients with peri-prosthetic fractures,
- Peri-synthesis fractures
- Pathological fractures, that is, on bones affected by primary tumor or metastasis, were excluded from the study.
- Likewise, patients who were referred to other hospitals without completing the treatment or follow-up period for any cause, except death.
After the new analysis, the result should be compared, point by point with the other signifiant reports.
Response: Thank you very much for pointing it out. The authors have carried out the comparison with other studies in the discussion.

This manuscript is a resubmission of an earlier submission. The following is a list of the peer review reports and author responses from that submission.
Round 1
Reviewer 1 Report
This paper studied risk factors affecting mortality at six months after hip fracture surgery.
All the perioperative variables have been statistically analyzed to find meaningful contributing factors in one university hospital of Spain.
- Abstract: L24, In the table and context of result you analyzed respiratory infection rather than respiratory failure. Why do you change the terminology to respiratory failure?
- Result: L139-143, You mentioned surgical delay and length of hospital stay increased mortality risk. Have you ever figured out why they are delayed? Are they hospital or patient related?
- Discussion: L296, What is the meaning of 'ecextus'? Is it mis-spell? There are so many studies in the literature in last 30 years about risk factor analysis for postop hip fracture mortality. What is the difference and uniqueness of this study from others? If you can say something adding to current understanding , it would be valuable and highly appreciated.
- Conclusion: They are the repetition of summary of result. It should be changed to more abbreviated form in order of significance under categorization.
Reviewer 2 Report
1. in table 1.1"ASA Surgical Risk" as well as "Surgical delay" and some others the Summ of the deaths are less than the expected (128), that can be in few cases but has to justified.
2. The reason of delayed surgery has to be justified (literature suggest to perform surgery as soon as it is possible, hopefully within 48 hours
3. the Nottingham Hip Fracture Score is not applied, but it could increase informations about the patients. The authors should give a short notice about why the didn't evaluate that.... or they should evaluate it.
4. the incidence of hip fracture (line 32-33 of the paper) in the population is claimed to be 104 cases per 100,000 inhabitants (it's low): further informations about aging index as well such a low incidence of hip fracture must be claryfied (suggestion: https://www.nature.com/articles/s41598-017-03847-x by Lucas)